# Large-Scale Heat-Tolerance Screening and Genetic Diversity of Pea (*Pisum sativum* L.) Germplasms

**DOI:** 10.3390/plants11192473

**Published:** 2022-09-21

**Authors:** Dong Wang, Tao Yang, Rong Liu, Nana Li, Naveed Ahmad, Guan Li, Yishan Ji, Chenyu Wang, Mengwei Li, Xin Yan, Hanfeng Ding, Xuxiao Zong

**Affiliations:** 1Institute of Crop Sciences, Chinese Academy of Agricultural Sciences, Beijing 100081, China; 2Institute of Crop Germplasm Resources, Shandong Academy of Agricultural Sciences, Jinan 250100, China; 3College of Life Science, Shandong Normal University, Jinan 250014, China

**Keywords:** pea (*Pisum sativum* L.), heat-tolerance screening, SNaPshot, genetic diversity, population genetic structure

## Abstract

Pea (*Pisum sativum* L.) is an important legume crop. However, the yield of pea is adversely affected by heat stress in China. In this study, heat-tolerant germplasms were screened and evaluated in the field under multi-conditions. The results showed that heat stress could significantly affect pea yield. On the basis of grain weight per plant, 257 heat-tolerant and 175 heat-sensitive accessions were obtained from the first year’s screening, and 26 extremely heat-tolerant and 19 extremely heat-sensitive accessions were finally obtained in this study. Based on SNaPshot technology, two sets of SNP markers, including 46 neutral and 20 heat-tolerance-related markers, were used to evaluate the genetic diversity and population genetic structure of the 432 pea accessions obtained from the first year’s screening. Genetic diversity analysis showed that the average polymorphic information content was lower using heat-tolerance-related markers than neutral markers because of the selective pressure under heat stress. In addition, population genetic structure analysis showed that neutral markers divided the 432 pea accessions into two subpopulations associated with sowing date type and geographical origin, while the heat-tolerance-related markers divided these germplasms into two subpopulations associated with heat tolerance and sowing date type. Overall, we present a comprehensive resource of heat-tolerant and heat-sensitive pea accessions through heat-tolerance screenings in multi-conditions, which could help genetic improvements of pea in the future.

## 1. Introduction

Pea (*Pisum sativum* L., 2n = 14) is a cool-season food legume crop widely grown in temperate regions around the world. It has high nutritional value and is an important source of protein, starch, sugar, crude fiber, vitamins, and low fat. Rhizobia in pea root nodules fix nitrogen in the atmosphere to increase soil fertility and reduce environmental pollution [1,2]. According to statistics of the Food and Agriculture Organization of the United Union (FAO), the global production of food legumes amounted to 88.4 million tons in 2019, with pea production ranking second after common beans (*Phaseolus vulgaris* L.), while the production of dry peas in China (1.5 million tons) ranked third and that of green peas (13.4 million tons) ranked first in the world [3]. However, China still imports a large amount of peas from Canada and other countries to meet the increasing consumption demand. Therefore, recent studies have increasingly focused on the genetics and varietal improvement of pea [4,5,6,7,8,9].

Global warming is increasingly affecting the production and yield of important crops. The global temperature has risen by 1.5 °C since the Industrial Revolution [10], and an increase of 1.5 to 2.0 °C is expected to be observed by the end of this century [11,12,13]. As a consequence, heat stress could be a major threat leading to plant yield loss, even in heat-resistant plants, such as sorghum [14]. Significant yield losses in cool-season food legumes, such as pea [15,16,17,18], chickpea (*Cicer arietinum* L.) [19,20], and lentil (*Lens culinaris* Medikus) [21], have been reported. Specifically, heat stress during flowering and seed development has been shown to be a significant contributor to such losses. Pea is particularly sensitive to high temperatures in the field, which may lead to the abortion of flowers, fruits, and seeds and reduction in seed size [22,23,24]. Under high-temperature conditions, heat-tolerant (HT) varieties produce more reproductive nodes per plant, fewer pods are lost per plant, and fewer seeds are lost per pod, while heat-sensitive (HS) varieties exhibit the opposite characteristics [25]. Consequently, understanding pea genetics and breeding HT pea varieties could have significant outcomes for coping with abiotic stresses in the future.

A thorough understanding of the genetic diversity and relationships among pea populations is necessary for pea genetic improvement research and the selection of appropriate parents in breeding programs [26]. A wide range of molecular markers have been previously used to evaluate the genetic diversity of pea germplasms [27,28,29,30,31,32,33,34,35,36,37,38,39]. Among these markers, single-nucleotide polymorphisms (SNPs) are third-generation molecular markers with potential advantages, such as bi-allelicity, abundant quantities, wide distribution, low mutation rates, and automatic high-throughput detection [40,41]. Recent advances in the utilization of SNP markers have made them extensively applicable in a variety of pea research areas, such as association mapping [42], genome-wide association studies (GWASs) [43], quantitative trait locus (QTL) identification [44], candidate gene mining [45], and genetic-linkage map construction [46]. Numerous approaches are available for performing SNP typing, including polymerase chain reaction–single-strand conformation polymorphism (PCR-SSCP), allele-specific PCR (AS-PCR) markers combined with polymerase chain reaction (PCR) technology and cleaved amplified polymorphic sequences (CAPS), next-generation sequencing (NGS), competitive allele-specific PCR (KASP), and mini-sequencing technology [41]. SNaPshot technology, also known as micro-sequencing technology, achieves mid-throughput SNP typing [47,48] and has been widely used in forensic medical identification [49] and in SNP detection of human genes [50] due to its high sensitivity, good repeatability, and lack of need for additional equipment. SNaPshot shows broad application prospects in the field of plant genetics research, such as SNP typing and marker development [51], molecular-marker-assisted breeding [52], and genetic diversity analysis [53,54,55,56]. To date, there have been only sporadic studies on the genetic diversity and population genetic structure of pea germplasms using SNP markers [57] and even fewer reports have analyzed pea genetics using SNaPshot technology.

At present, there have been only a few reports on the heat-tolerance screening of pea germplasms worldwide [25]. In particular, little is known about the genetic diversity of HT pea germplasms in China, which hinders the breeding and improvement of HT pea varieties. In this study, heat tolerance was evaluated for 2358 worldwide pea accessions in the field under three different sowing conditions over three years. A reliable classification standard for screening HT pea accessions was established through the identification of yield-related traits, and a total of 45 extremely HT and HS pea accessions were finally obtained. The genetic diversity and population genetic structure of the 432 pea accessions resulting from the first heat-tolerance screening were investigated using two sets of SNP markers identified through SNaPshot technology. This study provides theoretical and practical bases for in-depth understanding of the genetic mechanism underlying heat tolerance in pea, which will facilitate the development of new HT pea varieties in future breeding programs.

## 2. Results

### 2.1. Heat Stress during Heat-Tolerance Screenings

During the three heat-tolerance screenings, the daily average temperatures in the growth periods of pea at different sowing stages showed an increasing trend (Figure 1), and the daily average temperatures of the first stage of late sowing (LS1) and the second stage of late sowing (LS2) were significantly higher than those of normal sowing (NS). The optimal average daily temperature of the vegetative growth period was 12–16 °C, that of the flowering period was 16–20 °C, and that of the pod-setting period was 16–22 °C. Generally, when the temperature was higher than 26 °C, the metabolic activities of the pea plants tended to stop. The days when the average daily temperatures were higher than 16 °C, 20 °C, 22 °C, and 30 °C in the growth periods of pea during the three heat-tolerance screenings are shown in Figure 2. During each heat-tolerance screening, the growth periods (days) of pea accessions at NS, LS1, and LS2 decreased, while the numbers of days with average daily temperature higher than 16 °C, 20 °C, 22 °C, and 30 °C showed increasing trends. Based on these results, the effects of heat stress on pea accessions at LS1 and LS2 during the three heat-tolerance screenings were significant, meeting the requirements of the heat-tolerance screening experiments.

### 2.2. Field Survival Rate of Pea Accessions at Different Sowing Stages

The distribution of the field survival rates (FSRs) of the pea accessions at different sowing stags during the first heat-tolerance screening (HTS1) is shown in Figure 3A. The average FSR of the 2358 accessions at 2017NS was 61.8%, of which 1533 (65.0%) accessions demonstrated an FSR of ≥60%, and 401 (17.0%) accessions showed an FSR of 80%. The average FSR at 2017LS1 was 47.4%; the number of accessions having an FSR of ≥60% was 1011 (42.9%), which was significantly lower than that at 2017NS (*p* < 0.001), and the number of accessions with an FSR of zero was 322 (13.7%). The average FSR at 2017LS2 was 28.5%; the number of accessions having an FSR of ≥60% was 472 (20.0%), which was much lower than that at 2017NS (*p* < 0.001), and the number of accessions with an FSR of zero reached 733 (31.1%). The above results indicated that the FSRs of 2017LS1 and 2017LS2 were lower than that of 2017NS after heat-stress treatment, and the FSR at 2017LS2 was the lowest.

The average FSRs at 2018NS and 2019NS were 69.2% and 61.1%, respectively (Figure 3B,C). A total of 334 (77.3%) and 64 (59.3%) accessions at 2018NS and 2019NS, respectively, had an FSR of ≥60%. The average FSRs at 2018LS1 and 2019LS1 were 43.0% and 40.3%, respectively, of which 103 (23.8%) and 15 (13.9%) accessions, respectively, had an FSR of ≥60%, significantly lower than the corresponding values at 2018NS and 2019NS (*p* < 0.001). The average FSRs at 2018LS2 and 2019LS2 were 22.4% and 29.3%, respectively, of which 16 (3.7%) and 13 accessions (12.0%) had an FSR of ≥60%, much lower than the corresponding values at 2018NS and 2019NS (*p* < 0.001), respectively. These results showed that the FSRs of pea accessions at LS1 and LS2 were significantly lower than those at NS during the second heat-tolerance screening (HTS2) and the third heat-tolerance screening (HTS3), similar to the results obtained for HTS1.

### 2.3. Sowing Stages Affected the Grain Weight Per Plant

During HTS1, the average grain weight per plant of each pea accession was compared for different sowing stages (Figure 4A). In 2017NS, 661 (28.0%) accessions showed an average grain weight per plant of 0–10 g, while 189 (8.0%) accessions demonstrated an average grain weight per plant of >60 g and the accessions at other levels were evenly distributed. The number of accessions with an average grain weight per plant of 0–10 g was 959 (40.7%) at 2017LS1, which was higher than that at 2017NS. The number of accessions having other levels of average grain weight per plant was lower than that at 2017NS, indicating that heat stress exerted certain effects on the grain weights of pea accessions. The number of accessions with an average grain weight per plant of 0–10 g reached 1692 (71.8%) at 2017LS2, far exceeding the number of accessions with any category of grain weight per plant at 2017NS, indicating that heat stress exerted an extremely serious effect on the grain weights of pea accessions.

The distributions of average grain weight per plant of pea accessions during HTS2 and HTS3 are shown in Figure 4B,C, respectively. The numbers of accessions with an average grain weight per plant of 0–10 g at 2018NS and 2019NS were the largest, these being 182 (42.1%) and 39 (36.1%), respectively, and the number of accessions at other levels decreased regularly. In 2018LS1 and 2019LS1, the numbers of accessions with an average grain weight per plant of 0–10 g were 253 (58.6%) and 72 (66.7%), respectively, which figures were higher than those at 2018NS and 2019NS, whereas the numbers of accessions with an average grain weight per plant at other levels were lower than those at 2018NS and 2019NS. In 2018LS2 and 2019LS2, the numbers of accessions with an average grain weight per plant of 0–10 g were as high as 406 (94.0%) and 96 (88.9%), respectively, far exceeding the numbers of accessions with an average grain weight per plant of the corresponding and other categories at 2018NS and 2019NS. The above results confirmed that heat stress had a serious impact on the average grain weight per plant of pea accessions.

### 2.4. Evaluation of Pea Accessions According to Heat-Tolerance Levels

According to the classification standard for screening HT pea accessions, the number of accessions at each level after the three heat-tolerance screenings was determined, as shown in Figure 5. After HTS1, the number of accessions at each level showed an approximately normal distribution, among which 257 accessions were classified as HT, while 175 accessions were classified as HS.

HTS2 was conducted based on the above 432 HT and HS accessions obtained from HTS1. The number of accessions at each level showed a trend of polarization, that is, the proportions of HT (57) and HS (51) pea accessions were relatively increased, while the proportions of pea accessions at other levels were relatively decreased (Figure 5B). These 108 accessions (HT and HS) were used for HTS3.

After HTS3, the polarization of the accession fractions at each level was more obvious, that is, the numbers of HT (26) or HS accessions (19) were greater than those of pea accessions at the other levels (Figure 5C). The 26 HT and 19 HS accessions obtained after HTS3 were defined as extremely HT and extremely HS accessions, respectively.

The Kolmogorov–Smirnov test was used to assess the number of accessions at all levels after each heat-tolerance screening. The asymptotic significance (two-sided) probability (*p*) values were 0.850, 0.975, and 0.996, respectively, for HTS1, HTS2, and HTS3, which were all greater than 0.10, indicating that these data followed a normal distribution and that the three heat-tolerance screenings were reasonable and feasible.

### 2.5. Comparison of Yield Traits between HT and HS Pea Accessions

The average first-flowering dates and grain filling rates (GFRs) of the (extremely) HT and (extremely) HS accessions after HTS2 and HTS3 are shown in Appendix A. For a total of six sowing stages of HTS2 and HTS3, the average first-flowering dates were not significantly different between (extremely) HT and (extremely) HS accessions, ranging from 0 to 4 days. The difference in the average temperature was only 0.1–1.2 °C, and the average daily temperatures during the grain filling periods of the (extremely) HT accessions in 2018LS1, 2018LS2, and 2019LS1 were higher than those of the (extremely) HS accessions, indicating that the (extremely) HT accessions were tolerant to heat stress. Furthermore, this tolerance could not be ascribed to their escape from high temperature through early flowering but was due to their own characteristics, which confirmed the reliability of the heat-tolerance screening. No significant difference in GFRs was observed between the (extremely) HT and (extremely) HS accessions in 2018NS and 2019NS, indicating that temperature had no significant effect on grain filling of both types of pea accessions under normal sowing. In 2018LS1, 2018LS2, 2019LS1, and 2019LS2, the GFRs of (extremely) HT accessions were significantly higher than those of (extremely) HS accessions, and the GFRs of NS, LS1, and LS2 decreased gradually in both HS and HT pea accessions. The results indicated that the effect of heat stress on the grain filling of (extremely) HS accessions was much greater than for (extremely) HT accessions. Interestingly, the GFRs of the extremely HT accessions from HTS3 were higher than those of the HT accessions from HTS2, while the GFRs of the extremely HS accessions from HTS3 were lower than those of the HS accessions from HTS2, indicating that HTS3 deepened the screening effect.

### 2.6. Heat Tolerance and Sowing Date Type of Pea Accessions

According to sowing dates, the 2358 accessions were divided into spring-sowing (SS, 1324, 56.1%) and winter-sowing (WS, 1034, 43.9%) types (Table 1 and Appendix A). As described in Appendix A, among the 432 accessions obtained from HTS1, 246 (56.9%) were SS type and 186 (43.1%) were WS type. Among the 257 HT accessions, 100 (38.9%) were SS type and 157 (61.1%) were WS type. Of the 175 HS accessions, 146 (83.4%) were SS type and 29 (16.6%) were WS type. Among the 108 accessions obtained from HTS2, 80 (74.1%) were SS type and 28 (25.9%) were WS type. Among the 57 HT accessions, 35 (61.4%) were SS type and 22 (38.6%) were WS type. Of the 51 HS accessions, 45 (88.2%) were SS type and 6 (11.8%) were WS type. Among the 45 accessions obtained from HTS3, 29 (64.4%) were SS type and 16 (35.6%) were WS type. Among the 26 extreme HT accessions, 12 (46.2%) were SS type and 14 (53.8%) were WS type. Of the 19 extreme HS accessions, 17 (89.5%) were SS type and 2 (10.5%) were WS type. The above results showed that the numbers of SS-type pea accessions were smaller than those of WS-type pea accessions among the HT germplasms, while SS-type pea accessions were much more abundant than WS-type accessions in the HS germplasms.

### 2.7. Genetic Diversity of the 432 Pea Accessions from HTS1

The genetic diversity of the 432 pea accessions obtained from HTS1 was evaluated using two sets of SNaPshot markers. Using neutral markers, the total genotype number (NG) and allele number (NA) for these 432 accessions were 140 and 94, respectively, while the mean main allele frequency (MAF), gene diversity (GD), expected heterozygosity (He), and polymorphic information content (PIC) values were 0.705, 0.371, 0.155, and 0.293, respectively (Table 2), and the ranges were 0.505–0.988, 0.023–0.628, 0.005–0.539, and 0.023–0.577, respectively (Appendix A). According to PIC values, SNaPshot markers were divided into high-PIC (PIC ≥ 0.5), medium-PIC (0.25 ≤ PIC < 0.5), and low-PIC (PIC < 0.25) groups, with 1 high-PIC, 34 medium-PIC, and 11 low-PIC SNaPshot markers being identified (Table 2). Based on heat-tolerance-related markers, the total NG and NA for the 432 pea accessions were 52 and 39, respectively, while the mean MAF, GD, He, and PIC values were 0.749, 0.313, 0.156, and 0.246, respectively (Table 2), and the ranges were 0.530–1, 0–0.498, 0–0.488, and 0–0.374, respectively (Appendix A). Thirteen medium-PIC and 7 low-PIC SNaPshot markers were identified based on PIC values (Table 2). The results from the above analyses using two sets of SNaPshot markers revealed that the 432 pea accessions obtained from HTS1 had a significant degree of genetic variation.

### 2.8. Population Genetic Structure of the 432 Pea Accessions from HTS1

The genetic composition of the 432 pea accessions from HTS1 was calculated, and the optimal number of genetic subpopulations (K) was determined to study the population genetic structure of these pea accessions. For neutral markers, the highest Evanno’s ΔK value [58] was observed at K = 2 and was much higher than those observed at other K values (Appendix A). As shown in Figure 6A, Subpopulation A had 169 accessions, of which SS-type accessions constituted the major proportion (75.7%); 154 (91.1%) accessions were from Northern China, while 11 (6.5%) and 4 (2.4%) were from Southern China and regions outside China, respectively. Similarly, Subpopulation B contained 263 accessions, of which 118 (44.9%) were SS type and 145 (55.1%) were WS type; 111 (42.2%) accessions were from Southern China, 87 (33.1%) were from Northern China, 57 (21.7%) were from foreign countries, and 8 (3.0%) had unknown origins (Appendix A). Based on heat-tolerance-related markers, the highest Evanno’s ΔK value was also observed at K = 2 (Appendix A), and these pea accessions were accordingly divided into two subpopulations. Subpopulation C had 185 accessions, including 72 (38.9%) HT accessions and 113 (61.1%) HS accessions; among the 185 accessions, 126 (68.1%) were SS type and 59 (31.9%) were WS type (Figure 7A). Subpopulation D contained 247 accessions, including 185 (74.9%) HT accessions and 62 (25.1%) HS accessions; 120 (48.6%) accessions were SS type and 127 (51.4%) were WS type in Subpopulation D (Appendix A). Taken together, the two sets of SNaPshot markers divided the accessions into four subpopulations with substantial differences in the numbers and compositions of pea accessions.

The results of the structure analysis were further verified by principal coordinate analysis (PCoA). Based on neutral markers, PCoA divided the 432 pea accessions into two genetic subpopulations, A and B. As shown in Figure 6B, Subpopulation A was clearly separated from Subpopulation B, except that several accessions were within the other subpopulation. The population composition was consistent with that obtained from the genetic structure analysis. The contribution rate of the first three components of PCoA to neutral markers was 34.6%. Similarly, PCoA based on heat-tolerance-related markers also divided these pea accessions into two genetic subpopulations, C and D, consistent with the genetic structure analysis. As shown in Figure 7B, Subpopulation C, in the blue ellipse, was roughly separated from Subpopulation D, in the red ellipse. The contribution rate of the first three components of PCoA to heat-tolerance-related markers was 47.5%, which was higher than that of neutral markers. The above results indicated that the genetic subpopulation grouping of pea accessions from the structure analysis was well-supported by PCoA.

Phylogenetic trees were constructed by performing unweighted pair-group method with arithmetic means (UPGMA) clustering. Based on neutral markers, the 432 pea accessions were divided into two groups. As shown in Figure 6C, the red dendritic branches were classified as Subpopulation A and the green dendritic branches were classified as Subpopulation B. The UPGMA clustering based on heat-tolerance-related markers also divided these pea accessions into two groups. As shown in Figure 7C, the orange dendritic branches were classified as Subpopulation C, and the light blue dendritic branches were classified as Subpopulation D. Similar to the PCoA results, certain accessions were observed to cluster with those from the other subpopulations.

The 432 pea accessions were divided into SS type (*n* = 246) and WS type (*n* = 186), as well as HT accessions (*n* = 257) and HS accessions (*n* = 175), according to different indicators. Four subpopulations were obtained from the genetic structure analysis based on the two sets of SNaPshot markers and were used to analyze the genetic composition. As shown in Figure 8, among the 246 SS-type accessions, 128 (52.0%) belonged to Subpopulation A, slightly more than the 118 accessions (48.0%) in Subpopulation B. Similarly, among the 186 WS-type accessions, only 41 (22.0%) belonged to Subpopulation A, far fewer than the 145 (78.0%) in Subpopulation B. These findings indicated that more than half of the SS-type accessions belonged to Subpopulation A, while most WS-type accessions belonged to Subpopulation B. On the other hand, among the 175 HS accessions, 113 (64.6%) belonged to Subpopulation C, which was approximately twice the number (62, 35.4%) in Subpopulation D. On the contrary, among the 257 HT accessions, only 72 (28.0%) belonged to Subpopulation C, less than half of the 185 (72.0%) in Subpopulation D, indicating that most of the HS accessions belonged to Subpopulation C, while most of the HT accessions belonged to Subpopulation D.

## 3. Discussion

### 3.1. First Large-Scale Heat-Tolerance Screening of Pea Germplasms

Cool-season food legumes, including peas, broad beans (*Vicia faba* L.), chickpeas, and lentils, are important food sources for humans and are widely distributed around the world. However, these crops face adverse stresses, such as frost, heat, and drought, during their growth [59]. Studies on heat-tolerance screening of cool-season food legumes have previously been reported [60,61,62,63]. Pea germplasm resources have been studied in heat-tolerance screenings by assessing phenotypes in the field or performing laboratory experiments [25,64]. The present study is the first to screen heat tolerance in pea germplasms at a large scale (with up to 2358 accessions). Referring to the experimental methods of heat-tolerance screening in different cool-season food legumes, we adopted the following experimental design. First, data from different sowing stages were collected to ensure that the reproductive growth period of pea with a late sowing stage was consistent with that of high temperature. The late sowing period was divided into two stages, which better covered the responses of different pea genotypes to heat stress. Heat stress was used to screen the HT and HS pea accessions, which were shown to be particularly susceptible to heat stress during the flowering and pod-setting stages during pea pollen and ovary development. Second, a randomized block design was used to avoid the formation of a field microclimate, reduce systematic errors, and improve the accuracy of the experimental data. Third, heat tolerance was measured by determining the average grain weight loss rate per plant, because the biological response of crops to heat stress will ultimately reveal itself in yield attributes [25]. In addition, we also observed that different pea accessions showed different degrees of heat tolerance among the three heat-tolerance screenings. We finally identified 22 pea accessions showing consistent heat-tolerance results in the three screenings, of which 13 were extremely HT accessions (i.e., G0326, G0333, G0438, G0439, G0441, G0444, G0475, G0534, G1348, G1521, G2494, G3090, and G3094) and 9 were extremely HS accessions (i.e., G0272, G0586, G0679, G1036, G1078, G2051, G2507, G2494, and G2526). We recommend that the 13 extremely HT accessions can be used in future pea breeding programs to open a new direction in pea breeding and improve the heat tolerance of pea varieties.

Although a previous report has described the methodologies for screening and scoring cold tolerance in pea germplasms [7], the classification criteria for screening heat tolerance in pea germplasms have yet to be disclosed. In the present study, a heat-tolerance classification standard for pea accessions was generated, according to which the average grain weight loss rate per plant at the first (LR1) and second late-sowing stages (LR2) were used as key variables in the three heat-tolerance screenings. To eliminate systematic error, pea accessions with LR1 and LR2 values that differed by more than 40% were omitted from the study, ensuring the validity and accuracy of the results.

One of the breeding strategies for peas is to enhance their cold resistance, as peas are the main cool-season crops with a high degree of cold tolerance [8]. The proportion of cold-tolerant germplasms in WS areas is often higher than in SS areas. The reason is that pea germplasms in WS areas must survive in open-field environments, and the low-temperature stress is more severe than that in SS areas. Naturally, the proportion of cold-tolerant pea germplasms will increase after artificial breeding. As described in a previous study in which 3672 pea accessions were screened, 906 were collected from WS areas in Southern China, including 102 (11.3%) HR genotypes (consisting of green, healthy, and intact plants) and 318 (35.1%) MR genotypes (plants that survived but appeared yellow or brown). Among the 1411 pea accessions from SS areas in Northern China, 44 (3.1%) were HR genotypes and 245 (17.4%) were MR genotypes [7]. Consequently, it was validated that the proportion of cold-tolerant germplasms in WS areas was much greater than the proportion in SS areas. In this study, the proportion of WS-type HT accessions (84.4%) was much higher than that of SS-type HT accessions (40.7%), which was consistent with the aforementioned results regarding the proportions of cold-tolerant pea accessions of the two sowing date types. Based on these results, we speculate that a close relationship may exist between the heat-tolerance mechanism and the cold-tolerance mechanism in pea, which requires further in-depth research.

### 3.2. Analysis of the Genetic Diversity and Population Genetic Structure in Pea Germplasms Using SNaPshot Markers

SNaPshot technology is mainly used in forensic identification and SNP detection of human genes. In recent years, it has been applied in the field of plant genetics research due to its low cost, especially in research on forest trees and ornamental plants [51,52,54]. In the present study, the SNaPshot method was introduced for identification and evaluation of pea germplasms for the first time. The analyses of genetic diversity and population genetic structure of the 432 pea accessions obtained from HTS1 were conducted using two sets of SNaPshot markers, including 46 neutral markers and 20 markers related to heat-shock proteins or heat-shock transcription factors. The latter are closely related to heat tolerance in plants, including pea and cowpea [65,66]. Therefore, using two sets of SNaPshot markers allowed for the determination of correlations among heat-tolerance traits, sowing dates, and the geographical origins of pea germplasms.

After comparing the results obtained from the two sets of SNaPshot markers, we found that marker number significantly affected the total NG and NA of the pea germplasms and had a certain effect on the mean MAF, GD, and PIC values but almost no effect on mean He values. Noticeably, with increasing marker numbers, the total NG and NA increased, the mean MAF decreased, the mean GD and PIC increased, and the proportion of high- and medium-PIC markers increased and vice versa. Population size showed little effect on the total NG and NA, indicating that the two sets of markers were scientifically chosen and evenly distributed on the chromosomes. As the population size decreased, the mean MAF increased, the mean He did not change substantially, the mean GD and PIC decreased, and the proportion of high- and medium-PIC markers decreased and vice versa. In addition, the PIC of neutral markers was 0.293, which was higher than that of 20 markers related to heat-shock proteins or heat-shock transcription factors (0.246), indicating that the latter were under greater selective pressure under heat stress.

The neutral markers divided the 432 accessions into two genetic subpopulations. Subpopulation A had 169 accessions, of which 120 (71.0%) grown in Northern China belonged to the SS type. Subpopulation B contained 263 accessions, the top three categories of which were 99 (37.6%) WS-type accessions from Southern China, 60 (22.8%) SS-type accessions from Northern China, and 42 (16.0%) SS-type accessions from regions outside China, highly consistent with the actual pea production areas. Northern China belongs to the SS areas, while Southern China belongs to the WS areas. Most of the foreign accessions were from Europe and North America, which have higher latitudes and lower temperatures and thus are classified as SS areas. These results highlighted that pea accessions may be better classified according to their geographic origins and sowing date types using neutral markers, since these markers are not linked to functional genes, such as heat-tolerance-related genes.

Heat-tolerance-related markers also divided the 432 accessions into two genetic subpopulations. Subpopulation C contained 185 accessions, including 18 (9.7%) SS-type HT accessions, 54 (29.2%) WS-type HT accessions, 108 (58.4%) SS-type HS accessions, and 5 (2.7%) WS-type HS accessions. Subpopulation C was dominated by HS accessions, of which the SS-type accounted for the majority. Subpopulation D had 247 accessions, including 82 (33.2%) SS-type HT accessions, 103 (41.7%) WS-type HT accessions, 38 (15.4%) SS-type HS accessions, and 24 (9.7%) WS-type HS accessions. Subpopulation D was dominated by HT accessions, among which WS-type accessions accounted for a larger proportion. Several accessions were clustered with those from the other subpopulations. One possible explanation may be that the number of SNapshot markers used in this study was too small; increasing the number may resolve the ambiguous clustering. The results of this study were further verified; for example, a correlation was observed between heat tolerance and sowing date type in Subpopulations C and D. The explanation for this result may be that most of the SS areas have high latitudes, such as Liaoning, Inner Mongolia, northern Hebei, Shaanxi, Gansu, and Qinghai in Northern China, as well as those regions outside China (mainly Europe and North America); the temperature of these areas are seldom >30 °C during the whole growth period of pea. No natural selection pressure was observed on heat-tolerance genes in pea germplasms from the above areas, suggesting that these germplasms did not express heat-tolerance genes or contained fewer heat-tolerance gene copies. Therefore, they are very sensitive to heat stress. In pea production, WS-type pea varieties are usually sown in mid-to-late October and harvested in April to May of the following year. During the reproductive growth period, when the temperature is higher than 30 °C, the heat-tolerance genes are retained under selection pressure in WS-type pea varieties. Therefore, the heat tolerance of pea is associated with sowing date. In summary, our research corroborates that SNaPshot technology can be potentially utilized in pea breeding research to accelerate the mining of excellent genes in the future. To acquire more precise findings, more markers should be studied if the expense is affordable.

## 4. Materials and Methods

### 4.1. Plant Materials

A total of 2358 pea accessions were collected from the National Crop Genebank of China, Institute of Crop Sciences, Chinese Academy of Agricultural Sciences (Beijing, China). Among these accessions, 1973 (83.7%) were obtained from 28 provinces across China, 337 (14.3%) were from 25 countries and organizations outside China, and the rest (48 (2.0%)) had unknown origins. These pea accessions were divided into two types according to the sowing date: a spring-sowing (SS) type (1324, 56.1%) and a winter-sowing (WS) type (1034, 43.9%) (Table 1 and Appendix A).

### 4.2. Experimental Design for Heat-Tolerance Screenings

The three heat-tolerance screenings described in this study were conducted at the experimental farm in Nonggao District, Dongying City, Shandong Province (37.258614° N, 118.632774° E, altitude 14 m) in 2017; Wanlian Family Farm in Lingcheng District, Dezhou City, Shandong Province (37.472509° N, 116.590776° E, altitude 20 m) in 2018; and Jiyang District, Jinan City, Shandong Province (36.975931° N, 116.983151° E, altitude 21 m) in 2019, respectively.

For the first heat-tolerance screening (HTS1), a total of 2358 accessions were sown in three sowing conditions: normal sowing on 1 March 2017 (2017NS); first stage of late sowing on 16 March 2017 (2017LS1); and second stage of late sowing on 31 March 2017 (2017LS2). The purpose of conducting heat-tolerance screenings under late-sowing conditions (the latter two stages) was to apply heat stress in the reproductive growth stage of pea. Each sowing stage adopted a completely randomized design (CRD) and used artificial sowing. Each pea accession was sown with 10 seeds arranged in a row, with a row spacing of 0.5 m and a plant spacing of 0.1 m. Irrigation was performed before sowing to create moisture during seedling emergence. The experimental plot had sandy loam soil, and 600 kg of compound fertilizer (effective contents of N, P, and K ≥ 45%) was applied per hectare. At the emergence and flowering stages, 1.8% emulsifiable concentrate (5000×) abamectin was sprayed to control the leaf miner of pea, and weeds were removed manually. The numbers of surviving plants at the maturity stage for each pea accession in 2017NS, 2017LS1, and 2017LS2 were recorded.

For the second heat-tolerance screening (HTS2), the 432 accessions (heat-tolerant and heat-sensitive) obtained from HTS1 were sown under three sowing conditions, namely, normal sowing on 1 March 2018 (2018NS), the first stage of late sowing on 16 March 2018 (2018LS1), and the second stage of late sowing on 31 March 2018 (2018LS2). Different from HTS1, a randomized block design (RBD) with three replicates was used in HTS2 for each sowing stage to eliminate experimental errors. The first-flowering date, the flowering period, and the maturity period of pea accessions in each plot were investigated, and hundred-grain weight was recorded after harvest. The strategies of field design and management were consistent with those applied in HTS1. In the third heat-tolerance screening (HTS3), the 108 accessions (heat-tolerant and heat-sensitive) obtained from HTS2 were sown under three sowing conditions, namely, normal sowing on 1 March 2019 (2019NS), the first stage of the sowing period on 16 March 2019 (2019LS1), and the second stage of late sowing on 31 March 2019 (2019LS2). Each sowing stage adopted an RBD with three replicates, and the rest of the settings were the same as those for HTS2.

### 4.3. Meteorological Data Collection during Heat-Tolerance Screenings

The three experimental sites (i.e., Dongying, Dezhou, and Jinan) were located in the northeastern part of the North China Plain, with a temperate continental monsoon climate and four distinct seasons: hot and rainy in summer and cold and dry in winter. All meteorological data used during the experiments were downloaded from the website of the Shandong Meteorological Bureau (http://sd.cma.gov.cn/ (accessed on 11 January 2020)). Notably, 2017, 2018, and 2019 were normal years, the conditions ideal for distinguishing the HT and HS pea accessions.

The data for monthly average temperatures and monthly precipitation from March to June in Dongying, Dezhou, and Jinan were respectively recorded during the three heat-tolerance screenings (Appendix A). The thermal unit defined by Awasthi et al. was used to reflect the degree of heat stress in this study [67]. The ranges of variation in daily average temperature and thermal unit accumulation of the vegetative growth stage and reproductive growth stage for each sowing stage are shown in Appendix A. The temperature and thermal unit accumulation under late sowing in the three heat-tolerance screenings were significantly higher than those under normal sowing.

### 4.4. Classification Standard for Heat-Tolerance Screenings

For HTS1, five mature plants with pods and seeds were selected from each pea accession for the subsequnt analysis in 2017NS. The grain weight per plant for each plant was weighed, and the average grain weight per plant (W_NS_) was calculated for each accession. The same accessions were selected from 2017LS1 and 2017LS2, and their average grain weight per plant (W_LS1_ and W_LS2_) values were calculated. The average grain weight per plant values were also calculated for HTS2 and HTS3 in similar ways, only replicate experiments were set to eliminate errors.

The average grain weight loss rate per plant values for each accession in the first stage (LR1) and the second stage (LR2) of late sowing during the three heat-tolerance screenings were calculated using the following formulas:LR1 (%) = [1 − (W_LS1_/W_NS_)] × 100%
LR2 (%) = [1 − (W_LS2_/W_NS_)] × 100%

The criteria for classifying HT pea accessions were as follows:Level 1: 0 ≤ LR1 ≤ 20% and 0 ≤ LR2 ≤ 20%Level 2: (0 ≤ LR1 ≤ 20% and 20% < LR2 ≤ 40%) or (20% < LR1 ≤ 40% and 0 ≤ LR2 ≤ 20%)Level 3: 20% < LR1 ≤ 40% and 20% < LR2 ≤ 40%Level 4: (20% < LR1 ≤ 40% and 40% < LR2 ≤ 60%) or (40% < LR1 ≤ 60% and 20% < LR2 ≤ 4 0%)Level 5: 40% < LR1 ≤ 60% and 40% < LR2 ≤ 60%Level 6: (40% < LR1 ≤ 60% and 60% < LR2 ≤ 80%) or (60% < LR1 ≤ 80% and 40% < LR2 ≤ 60%)Level 7: 60% < LR1 ≤ 80% and 60% < LR2 ≤ 80%Level 8: (60% < LR1 ≤ 80% and 80% < LR2 ≤ 100%) or (80% < LR1 ≤ 100% and 60% < LR2 ≤ 80%)Level 9: 80% < LR1 ≤ 100% and 80% < LR2 ≤100%.

If the difference between LR1 and LR2 was greater than 40% for an accession, this accession was excluded from the classification. Pea accessions of levels 1 to 3 were considered HT, and those of levels 7 to 9 were considered HS.

### 4.5. SNaPshot Analysis

The 432 pea accessions obtained from HTS1 were used for SNaPshot analysis. For each accession, young leaves of 3 plants were collected after 4 weeks of sowing in 2018NS and mixed. Genomic DNA (gDNA) was extracted using a TSINGKE Plant DNA Extraction Kit (Tsingke Biotechnology Co., Ltd., Beijing, China). The design of peripheral primers was based on the following principles: the length of the primer was 15–30 bp and its effective length was ≤38 bp; the GC content should range from 40% to 60%; the optimum Tm value was kept between 58 and 60 °C; the primer itself must not contain self-complementary sequences; ≤4 complementary or homologous bases should be present between the primers; and complementary overlaps at the 3’ end should be avoided. The design principles for single-base extension primers were as follows: the primer length was 15–30 bp, the GC content was 40%–60%, and the optimal Tm value was 58–60 °C. Different lengths of PolyC or PolyT were added to the 5′ ends of the primers, such that each primer could be distinguished by length. The shortest primer, after tailing, was 36 bp, and the length of the primers at two adjacent SNP sites generally differed by 4–6 nucleotides. The GenoPea 13.2K SNP chip developed by Tayeh et al. was used for SNaPshot analysis [68]. Two sets of SNP loci were selected, including 46 loci with neutral mutations and 20 loci related to heat-shock proteins or heat-shock transcription factors. For each SNP locus, Primer Premier v5 software was used to design a pair of peripheral primers and a single-base extension primer. The information on SNP loci and SNaPshot primers is shown in Appendix A.

The extracted DNA sample was diluted to 20 ng/μL. The 35 μL PCR system contained 30 μL of 1.1× T3 Super PCR Mix (Tsingke Biotechnology Co., Ltd., Beijing, China), 2 μL of 10 μM forward primer, 2 μL of 10 μM reverse primer, and 1 μL of template (gDNA). The PCR procedures were 98 °C for 3 min; 35 cycles of 98 °C for 10 s, 57 °C for 10 s, and 72 °C for 15 s; 72 °C for 2 min; and 4 °C storage. The PCR products were subjected to agarose gel electrophoresis, and the target band size was determined using a gel imager. The PCR products purified using a MagS magnetic bead gel recovery kit (Tsingke Biotechnology Co., Ltd., Beijing, China) were ready for use, and the single-base extension primer was diluted to 10 μM for SNaPshot PCR. The SNaPshot PCR system had a 5 μL volume, which contained 2 μL of ABI SnapShot multiplex Mix (Applied Biosystems, Foster City, CA, USA), 1 μL of primer, 1 μL of purified PCR template, and 1 μL of ddH2O. The amplification procedures were 96 °C for 2 min; 30 cycles of 96 °C for 10 s, 50 °C for 5 s, 60 °C for 30 s; 60 °C for 30 s; and 4 °C storage. SNaPshot PCR products were detected by capillary electrophoresis using an ABI 3730 × L DNA Analyzer (Applied Biosystems, Foster City, CA, USA).

### 4.6. Data Analysis

Excel was used to calculate the field survival rate (FSR) of each pea accession in different sowing stages with the following formula:(1)$$\mathrm{Field}\;\mathrm{survival}\;\mathrm{rate}\;(\%)=\left[\overset{n}{\underset{i=1}{\sum }}\left(\frac{\mathrm{Number}\;\mathrm{of}\;\mathrm{surviving}\;\mathrm{plants}\;\mathrm{at}\;\mathrm{the}\;\mathrm{mature}\;\mathrm{stage}}{10}\times 100\%\right)\right]/3$$
where n is the number of repetitions set in the experiment, with n = 1 for HTS1 and n = 3 for HST2 and HST3.

The grain filling rates (GFRs) of each pea accession in different sowing stages for HST2 and HST3 were calculated in Excel using the following formula:(2)$$\mathrm{Grain}\;\mathrm{filling}\;\mathrm{rate}\;\left(g/d\right)=\frac{\mathrm{Hundred}\;\mathrm{grain}\;\mathrm{weight}\;\left(g\right)}{\mathrm{Maturity}\;\mathrm{period}\;\left(d\right)-\mathrm{Flowering}\;\mathrm{period}\;\left(d\right)}$$

Analysis of variance (ANOVA) and mean comparisons were conducted on GFRs between HT and HS accessions at different sowing stages, and the significance of differences was tested.

GeneMapper v4.1 was utilized to analyze the SNP loci. Genotyping was performed for each sample according to the corresponding peak value of SNP loci. PowerMarker v3.25 [69] was used to calculate the genetic diversity indices of pea germplasms based on the two sets of SNP markers, including genotype number (NG), main allele frequency (MAF), allele number (NA), genetic diversity (GD), expected heterozygosity (He), and polymorphic information content (PIC).

The two sets of SNP markers were utilized to examine the genetic structure of the 432 pea accessions obtained from HTS1. Firstly, STRUCTURE v2.3.4 was used for a Bayesian clustering analysis [70,71] with the following parameters: number of burn-ins = 10,000, number of MCMC repetitions after burn-in = 100,000, number of populations = 1–10, and number of iterations = 10. The optimal group structure and group number were determined by calculating the ΔK values (http://taylor0.biology.ucla.edu/struct_harvest/ (accessed on 23 February 2020)), according to Eral et al. [72]. Secondly, GenAlEx v6.5 [73] was used for principal coordinate analysis (PCoA) to assess whether the population structures of pea germplasms obtained from heat-tolerance screenings were reasonable. Finally, PowerMarker v3.25 [69] was used to construct a phylogenetic tree via the unweighted pair-group method with arithmetic means (UPGMA) for the 432 pea accessions, which was visualized by Exhibited v1.4.3 (https://github.com/rambaut/figtree/releases/tag/v1.4.3 (accessed on 6 March 2020)). In addition, significance tests of the mean differences and Kolmogorov–Smirnov tests were completed using SPSS v20.0.

## 5. Conclusions

A heat-tolerance screening of 2358 pea accessions collected from different regions of the world was conducted in this study. After HTS1, 257 HT accessions and 175 HS accessions were obtained, and 26 extremely HT accessions and 19 extremely HS accessions were obtained after HTS3. In addition, a corresponding heat-tolerance classification standard was formulated. Most HT accessions were WS type, while the HS accessions included more SS-type than WS-type accessions. Two sets of SNaPshot markers were used to analyze the genetic diversity and population genetic structure of the 432 pea accessions obtained from HTS1. The purpose was to determine the correlations among heat-tolerance traits, sowing dates, and the geographical origins of pea germplasms using two sets of SNaPshot markers. The genetic diversities of the pea accessions based on neutral markers were slightly higher than those based on heat-tolerance-related markers, indicating that heat-tolerance-related markers may have been affected by selection pressures under heat stress. In addition, pea sowing date type was found to be correlated directly with geographic origins on the basis of neutral markers, whereas pea heat tolerance was shown to be primarily associated with sowing date type on the basis of heat-tolerance-related markers. We recommend 22 pea accessions, including 13 extremely HT accessions and 9 extremely HS accessions, which can be used as parents for breeding HT pea varieties and conducting heat-tolerance gene mining in the future.

## Figures and Tables

**Figure 1 plants-11-02473-f001:**
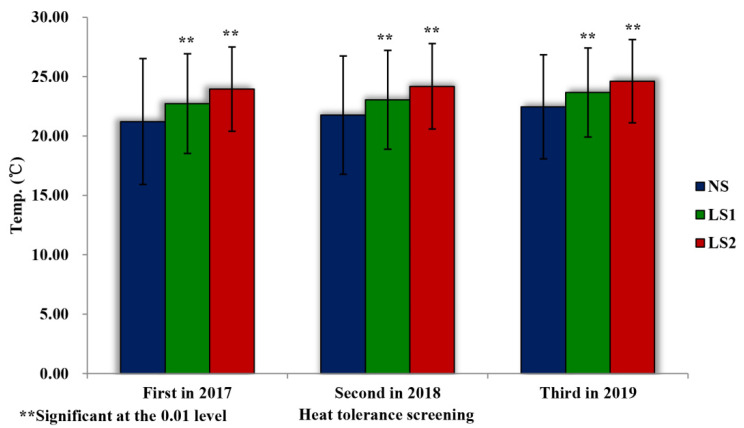
Comparison of daily average temperature changes in the growth periods of pea accessions at different sowing stages during the three heat-tolerance screenings.

**Figure 2 plants-11-02473-f002:**
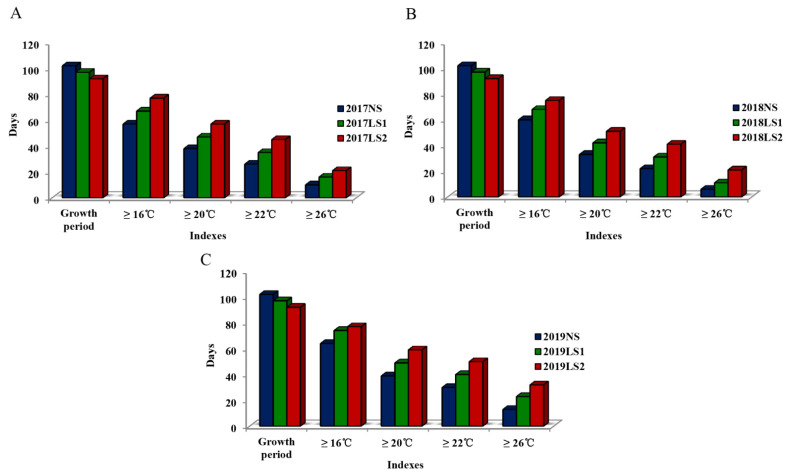
Comparison of the number of heat-stress days in the growth periods of pea accessions at different sowing stages during the three heat-tolerance screenings. (**A**) HST1 in 2017. (**B**) HST2 in 2018. (**C**) HST3 in 2019.

**Figure 3 plants-11-02473-f003:**
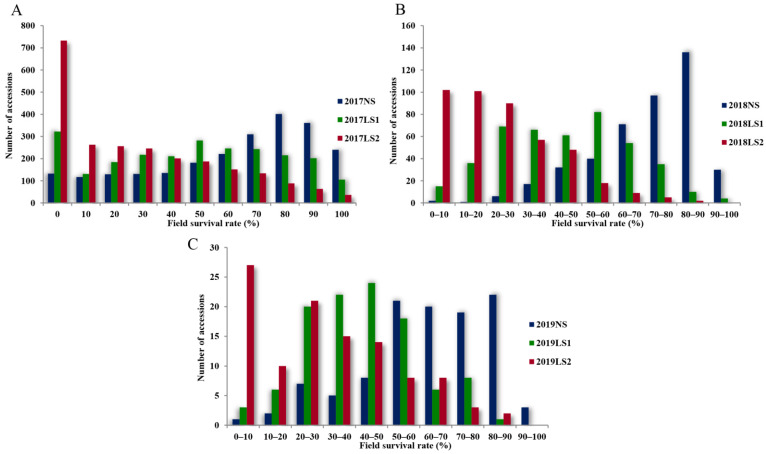
Distribution of FSRs of pea accessions at different sowing stages during the three heat-tolerance screenings. (**A**) HST1 in 2017. (**B**) HST2 in 2018. (**C**) HST3 in 2019.

**Figure 4 plants-11-02473-f004:**
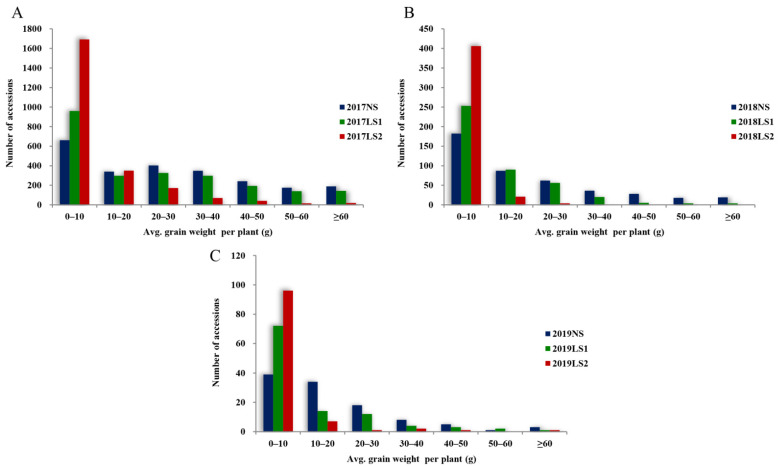
Distribution of average grain weight per plant of each pea accession after the three heat-tolerance screenings. (**A**) HST1 in 2017. (**B**) HST2 in 2018. (**C**) HST3 in 2019.

**Figure 5 plants-11-02473-f005:**
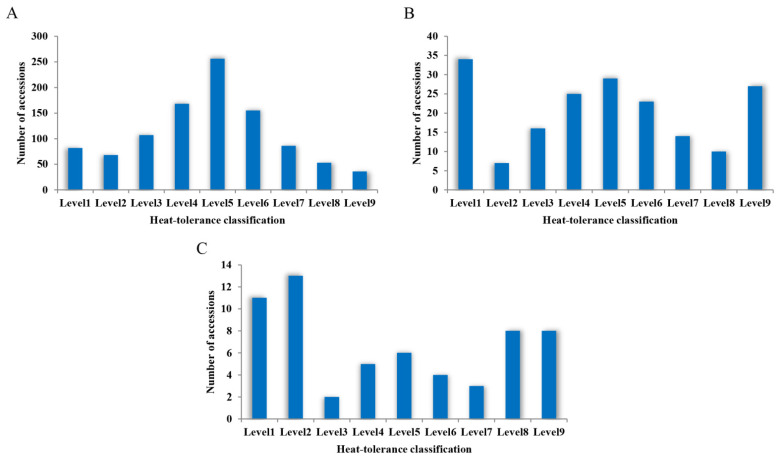
Distribution of the number of pea accessions at each level after the three heat-tolerance screenings. (**A**) HST1 in 2017. (**B**) HST2 in 2018. (**C**) HST3 in 2019.

**Figure 6 plants-11-02473-f006:**
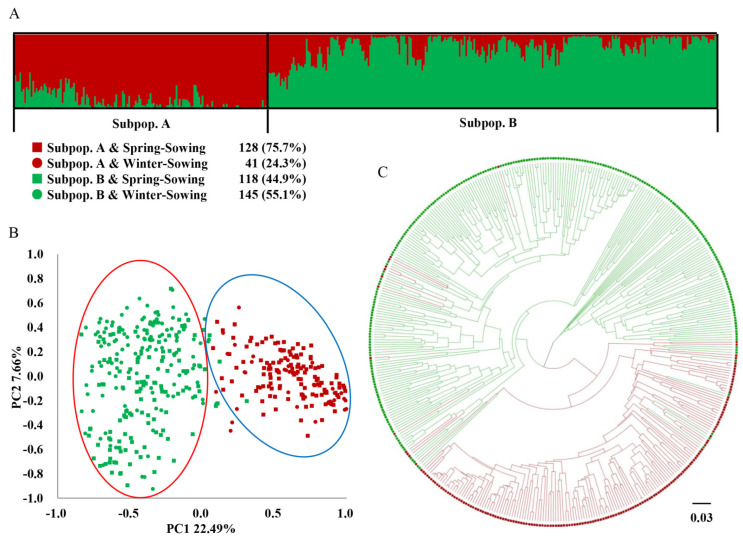
Population genetic structure of 432 pea accessions after HTS1 using neutral SNaPshot markers. (**A**) Structure analysis with 46 neutral SNaPshot markers. (**B**) PCoA with 46 neutral SNaPshot markers: the dark red squares represent the SS-type accessions of Subpopulation A and the dark red circles represent the WS-type accessions of Subpopulation A; the green squares represent the SS-type accessions of Subpopulation B and the green circles represent the WS-type accessions of Subpopulation B. (**C**) UPGMA tree based on Nei’s genetic distance with 46 neutral SNaPshot markers.

**Figure 7 plants-11-02473-f007:**
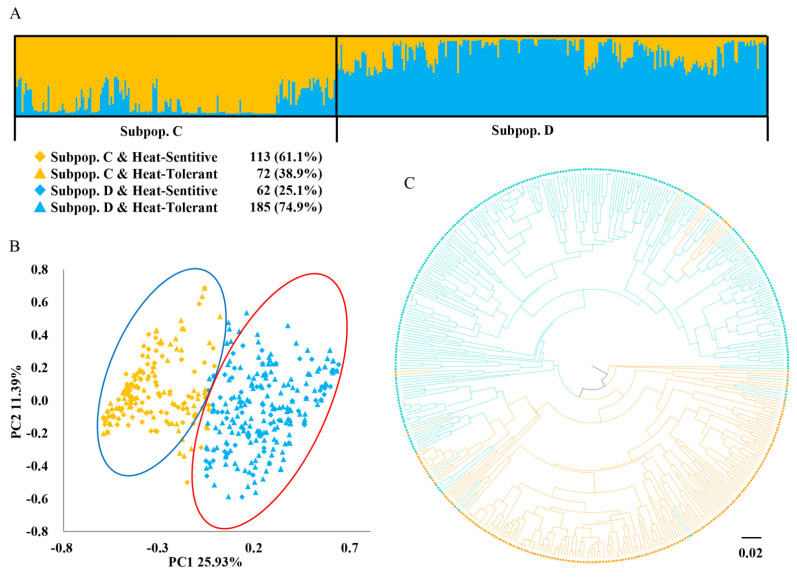
Population genetic structure of 432 pea accessions after HTS1 using heat-tolerance-related SNaPshot markers. (**A**) Structure analysis with 20 heat-tolerance-related SNaPshot markers. (**B**) PCoA with 20 heat-tolerance-related SNaPshot markers: the orange diamonds represent the SS type of Subpopulation C and the orange triangles represent the WS type of Subpopulation C; the light blue diamonds represent the SS type of Subpopulation D and the light blue triangles represent the WS type of Subpopulation D. (**C**) UPGMA tree based on Nei’s genetic distance with 20 heat-tolerance-related SNaPshot markers.

**Figure 8 plants-11-02473-f008:**
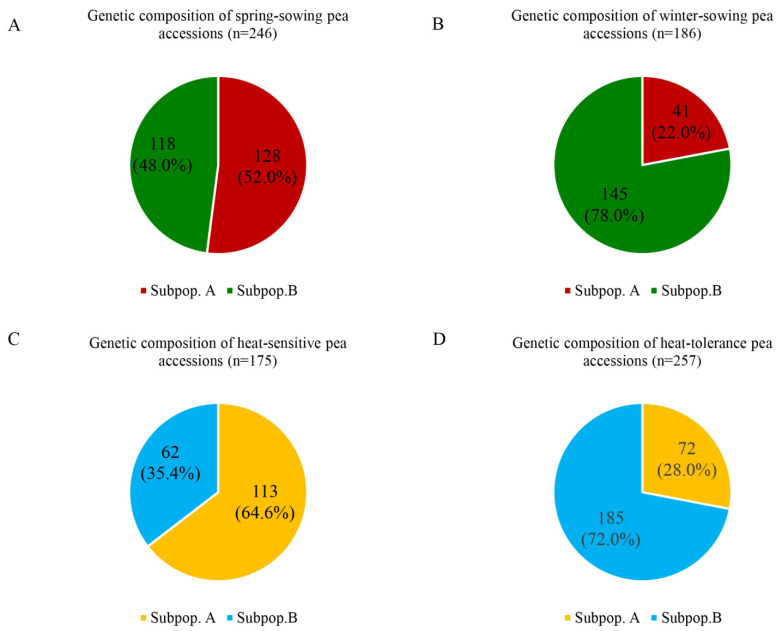
Population genetic structure of 432 pea accessions after HTS1. Genetic composition of (**A**) spring-sowing pea accessions (*n* = 246) and (**B**) winter-sowing pea accessions (*n* = 186), with 46 neutral SNaPshot markers. Genetic composition of (**C**) heat-sensitive pea accessions (*n* = 175) and (**D**) heat-tolerant pea accessions (*n* = 257), with 20 heat-tolerance-related SNaPshot markers.

**Table 1 plants-11-02473-t001:** Origin and sowing date type of 2358 pea accessions.

Origin	Accession Number	Sowing Date Type	
		Spring-Sowing	Winter-Sowing
Shaanxi, China	257	5	252
Inner Mongolia, China	237	236	1
Qinghai, China	178	178	
Hubei, China	174	8	166
Sichuan, China	173	4	169
Shanxi, China	144	143	1
Gansu, China	117	113	4
Xinjiang, China	103	98	5
Henan, China	90	48	42
Guizhou, China	78	6	72
Anhui, China	75	7	68
Chongqing, China	65		65
Yunnan, China	49	21	28
Tibet, China	41	41	
Guangxi, China	37	1	36
Jiangxi, China	28	1	27
Ningxia, China	25	25	
Jiangsu, China	20	3	17
Hunan, China	18	2	16
Liaoning, China	17	16	1
Shanghai, China	12	12	
Hebei, China	11	11	
Beijing, China	8	5	3
Guangdong, China	8	4	4
Taiwan, China	3	3	
Zhejiang, China	2	1	1
Fujian, China	1	1	
Heilongjiang, China	1	1	
Shandong, China	1	1	
Domestic Total	1973	995	978
United States	128	106	22
Germany	46	45	1
United Kingdom	28	17	11
Nepal	19	18	1
Bulgaria	13	13	
France	11	10	1
IGARDA	10	10	
Japan	10	8	2
Syria	9	8	1
Canada	7	7	
Russian Federation	8	7	1
Hungary	7	7	
New Zealand	6	6	
Australia	5	4	1
Poland	5	5	
Czech	5	5	
Turkey	5	4	1
India	4	4	
Denmark	3	3	
Chile	3	3	
Egypt	1	1	
Ethiopia	1	1	
Netherlands	1		1
Sudan	1		1
Greece	1		1
Foreign Total	337	292	45
Unknown	48	37	11
Spring-Sowing Total		1324	
Winter-Sowing Total			1034
Total	2358		

**Table 2 plants-11-02473-t002:** Summary of SNP marker genetic diversity parameters of 432 pea accessions after HTS1.

Marker Group	Marker Number	Total NG	Total NA	Mean MAF	Mean GD	Mean He	Mean PIC	Informative Type		
								Slight	Moderate	High
Ⅰ	46	140	94	0.705	0.371	0.155	0.293	11 (23.9%)	34 (73.9%)	1 (2.2%)
Ⅱ	20	52	39	0.749	0.313	0.156	0.246	7 (35.0%)	13 (65.0%)	0

Markers were classified as highly informative (PIC ≥ 0.5), moderately informative (0.25 ≤ PIC < 0.5), and slightly informative (PIC < 0.25). NG, genotype number; NA, allele number; MAF, major allele frequency; GD, gene diversity; He, heterozygosity; PIC, polymorphic information content.

## Data Availability

Not applicable.

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
