# Peer review of "Large-Scale Heat-Tolerance Screening and Genetic Diversity of Pea (Pisum sativum L.) Germplasms"

_plants, 2022, doi:10.3390/plants11192473_

Round 1

Reviewer 1 Report

The manuscript reported the genetic diversity of pea germplasm for heat-stress survey and SNP analysis. Novel heat-tolerant accessions for potential breeding purpose were indicated. The result is interesting for the genetic resources conservation of pea species. The study was clearly presented and the conclusions were supported by multiple evidences.

 This manuscript should be suitable for publication after replying below questions and considering revision:

1. Check overall the manuscript, the scientific name of species must be italic.

2. The text in each figure was not clear enough, and they should be edited for higher resolution.

3. Lane 586, the part of “Classification standard for heat-tolerance screenings”, the relevant references or literature to support the present classified parameters may be provided, which can be accepted by other researchers in the field.

Author Response

Responses to Reviewers 1

  1. Check overall the manuscript, the scientific name of species must be italic.

Response: Yes, your concern is reasonable. We have checked overall the manuscript, and all species name of species have been revised to italics, including the references.

------------------------------------------------

  1. The text in each figure was not clear enough, and they should be edited for higher resolution.

Response: As suggested, we increased the pixels in the picture, and all figures are at least 300pi to ensure their resolution.

------------------------------------------------

  1. Line 586, the part of “Classification standard for heat-tolerance screenings”, the relevant references or literature to support the present classified parameters may be provided, which can be accepted by other researchers in the field.

Response: Yes, your concern is reasonable. However, there are few literatures on large-scale screening of pea heat tolerance currently. Our experimental design is mainly based on the strategies and experiences of the previous researches on frost tolerance screening of pea germplasms,which has been repeatedly verified and the results are reliable (Zhang et al., Large-scale evaluation of pea (Pisum sativum L.) germplasm for cold tolerance in the field during winter in Qingdao. Crop J. 2016, 4, 377–383; Liu et al., Marker-trait association analysis of frost tolerance of 672 worldwide pea (Pisum sativum L.) collections. Sci. Rep. 2017, 7(1), 5919.). Besides, in order to meet the experimental goals and obtain reliable experimental results, we also made some appropriate adjustments. We tried to divide the experiment in each year into three sowing periods to ensure the reliability of the screening results, and the results were in line with our expectations. Therefore, we hope our methods could provide references for other colleagues.

------------------------------------------------

Reviewer 2 Report

A very detailed and extensive study.

Author Response

Responses to Reviewers 2

A very detailed and extensive study.

Response: Thank you very much for your affirmation. This study has indeed been carefully designed. In this study, heat tolerance germplasms were screened and evaluated in the field under multi-conditions. The results showed that heat stress could significantly affect pea yield. Through the grain weight per plant, 257 heat-tolerant and 175 heat-sensitive accessions were obtained from the first year’s screening, and 26 extremely heat-tolerant and 19 extremely heat-sensitive accessions were finally obtained in this study. Based on SNaPshot technology, two sets of SNP markers, including 46 neutral and 20 heat-tolerance-related markers, were used to evaluate the genetic diversity and population genetic structure of the 432 pea accessions obtained from the first year’s screening. Genetic diversity analysis showed that the average polymorphic information content was lower using heat-tolerance-related markers than neutral markers because of the selective pressure under heat stress. In addition, population genetic structure analysis showed that neutral markers divided the 432 pea accessions into two sub-populations associated with sowing date type and geographical origin. While the heat-tolerance-related markers divided these germplasms into two sub-populations associated with heat tolerance and sowing date type. Taken together, we present a comprehensive resource of heat-tolerant and heat-sensitive pea accessions through heat-tolerance screenings in multi-conditions, which would help genetic improvements of pea in the future.

------------------------------------------------

Reviewer 3 Report

 In this research, the authors used more than 2000 pea germplasm resources worldwide to evaluate the heat tolerance at different sowing conditions for many years, which was a large and meaningful work. Furthermore, two sets of SNP markers were used to classify the geographical origin and heat tolerance of 432 pea accessions. This work lays a great material foundation for the creation of pea germplasm resources and the research of molecular mechanism of pea heat tolerance in the future. But there are still some concerns to be raised:

1)  The author uses a large number of abbreviations, and only gives comments at the beginning of the article, which makes it inconvenient to read. The author should annotate the full vocabulary at the first time the text abbreviations appear.

2)  Please place the letter labels of Figure S2 and S3 on the upper left for easy reading.

3)Is it more accurate to choose an environment to carry out the test for three years?

Was there a great difference in temperature between three locations?  and how to ensure that there was high temperature influence in each locations?

4)are there any other index could be used except for HT and HS?

5)What is the purpose of structure analysis? please rewrite this content in the discussion and conclusion. 

6) Most HT accessions were WS-type, while HS accessions included more SS-type than WS-type accessions. This is a new conclusion that has not been reported before?

7) Are LR1 and LR2 the most reliable indicators? is there any relevant literature?

8) two sets of SNP markers, including 46 neutral and 20 heat-tolerance-related markers,  please provide the reasons why select these markers in the materials and methods. 

9) many contents are repeated in the results, discussion and conclusion part. please check it. 

Author Response

Responses to Reviewers 3

  1. The author uses a large number of abbreviations, and only gives comments at the beginning of the article, which makes it inconvenient to read. The author should annotate the full vocabulary at the first time the text abbreviations appear.

Response: Well, your concern is reasonable. We have annotated the full vocabulary at the first time the text abbreviations appear. In fact, the full vocabulary of abbreviations appeared in Materials and Methods.

------------------------------------------------

  1. Please place the letter labels of Figure S2 and S3 on the upper left for easy reading.

Response: Yes, your concern is reasonable. We have placed the letter labels of Figure S2 and S3 on the upper left, and please see the attached picture for details.

------------------------------------------------

  1. Is it more accurate to choose an environment to carry out the test for three years? Was there a great difference in temperature between three locations? And how to ensure that there was high temperature influence in each locations?

Response: Thanks for your comments. Please let us explain that the choice of three experimental sites is to verify the geographical adaptability of the pea germplasms for heat tolerance screening, that is, three experimental sites in three years are more convincing than one experimental site in three years. There are little difference in temperature trend among them, and the specific temperature value can refer to Figure S1. In fact, three locations have a temperate continental climate, with hot and rainy summers and cold, dry winters. Therefore, all three locations are guaranteed to have high temperatures for heat-tolerant screening of pea germplasms.

------------------------------------------------

  1. Are there any other index could be used except for HT and HS?

Response: Thanks for your comments. Actually, in order to keep consistency with other studies, we think there are not any other more suitable index except for HT and HS at present. Currently, there are few literatures on large-scale screening of pea heat tolerance. Our experimental design is mainly based on the strategies and experiences of our previous researches on frost tolerance screening of pea germplasms (Zhang et al., 2016. Large-scale evaluation of pea (Pisum sativum L.) germplasm for cold tolerance in the field during winter in Qingdao. Crop J. 4, 377–383; Liu et al., 2017. Marker-trait association analysis of frost tolerance of 672 worldwide pea (Pisum sativum L.) collections. Sci. Rep. 7(1), 5919.) as well as heat tolerance studies of other crop germplasm. In the study of heat tolerance of other crop germplasm, germplasm resources are usually divided into two categories: heat-tolerant (HT) and heat-sensitive (HS), which have been reported in several studies (Lucas et al., 2013. Markers for breeding heat-tolerant cowpea. Mol. Breeding 31: 529–536.; Sita et al., 2017. Identification of High-Temperature Tolerant Lentil (Lens culinaris Medik.) Genotypes through Leaf and Pollen Traits. Front. Plant Sci. 8, 744.; Chaudhary et al., 2020. Identification and characterization of contrasting genotypes/cultivars for developing heat tolerance in agricultural crops: current status and prospects. Front. Plant Sci. 11, 587264.).

------------------------------------------------

  1. What is the purpose of structure analysis? Please rewrite this content in the discussion and conclusion.

Response: Thank you very much for your suggestion. The purpose of structure analysis is to construct the correlation among heat tolerance trait, sowing date type and the geographical origin of pea germplasms using two sets of SNaPshot markers. We have rewritten this content in the discussion (lines 478-485, 496-499) and conclusion (lines 719-721). Please refer to the revised manuscript (Wang et al-manuscript-revision-edited) for details.

------------------------------------------------

  1. Define all the abbreviations Most HT accessions were WS-type, while HS accessions included more SS-type than WS-type accessions. This is a new conclusion that has not been reported before?

Response: Yes, we have defined all the abbreviations as suggested. For the concern on “Most HT accessions were WS-type, while HS accessions included more SS-type than WS-type accessions.”, this is a new conclusion we found in this study, which has not been reported before. This is also one of the innovative points of this study.

------------------------------------------------

  1. Are LR1 and LR2 the most reliable indicators? Is there any relevant literature?

Response: Thank you for your comments. Because the heat tolerance of pea germplasm is ultimately reflected in the reduction of grain weight per plant, the average loss rate of grain weight per plant was used as the index to measure the heat tolerance of pea germplasm, which was based on the research results of cold tolerance screening of pea germplasm. The references are the same as Question 4.

------------------------------------------------

  1. Two sets of SNP markers, including 46 neutral and 20 heat-tolerance-related markers, please provide the reasons why select these markers in the materials and methods.

Response: Thank you for your comments. “Two sets of SNP markers, including 46 neutral and 20 heat-tolerance-related markers” come from the GenoPea 13.2K SNP chip which was developed by Tayeh et al. (Tayeh et al., 2015. Development of two major resources for pea genomics: the GenoPea 13.2K SNP Array and a high-density, high-resolution consensus genetic map. Plant J. 84, 1257–1273.), and two groups of SNP loci were selected. Among them, forty-six loci in Group I were all neutral mutations, and twenty loci in Group II were related to heat shock proteins or heat shock transcription factors. Moreover, the selected SNP loci are evenly distributed on the pea chromosome. For each SNP locus sequence, Premier 5 software was used to design a pair of peripheral amplification primers and a single base extension primer. SNP loci and SNaPshot primer information are shown in Table S3 and Table S4.

------------------------------------------------

  1. Many contents are repeated in the results, discussion and conclusion part. Please check it.

Response: Thank you for your suggestion. We have checked it and deleted duplicate parts in the results, discussion and conclusion (lines 445-447, 508-509, 521-522 723-728). Please refer to the revised manuscript (Wang et al-manuscript-revision-edited) for details.

------------------------------------------------

Round 2

Reviewer 3 Report

Maybe the manuscript could be accept in current form.